# Structure of the SAS-6 cartwheel hub from *Leishmania major*

Mark van Breugel*, Rainer Wilcken[†], Stephen H McLaughlin, Trevor J Rutherford, Christopher M Johnson

Medical Research Council Laboratory of Molecular Biology, Cambridge, United Kingdom

**Abstract** Centrioles are cylindrical cell organelles with a ninefold symmetric peripheral microtubule array that is essential to template cilia and flagella. They are built around a central cartwheel assembly that is organized through homo-oligomerization of the centriolar protein SAS-6, but whether SAS-6 self-assembly can dictate cartwheel and thereby centriole symmetry is unclear. Here we show that *Leishmania major* SAS-6 crystallizes as a 9-fold symmetric cartwheel and provide the X-ray structure of this assembly at a resolution of 3.5 Å. We furthermore demonstrate that oligomerization of *Leishmania* SAS-6 can be inhibited by a small molecule in vitro and provide indications for its binding site. Our results firmly establish that SAS-6 can impose cartwheel symmetry on its own and indicate how this process might occur mechanistically in vivo. Importantly, our data also provide a proof-of-principle that inhibition of SAS-6 oligomerization by small molecules is feasible.

## Introduction

Eukaryotic flagella and cilia have essential roles in cell locomotion, fluid movement, and sensing. Their formation strictly requires the presence of centrioles, cylindrical organelles with a radially ninefold symmetric, peripheral microtubule array (*Bettencourt-Dias and Glover, 2007*; *Azimzadeh and Marshall, 2010*; *Gonczy, 2012*). During ciliogenesis/flagellogenesis, centrioles (then referred to as basal bodies) dock to the cell membrane via their distal ends and nucleate the flagellar/ciliar axoneme from their microtubule array (*Sherwin and Gull, 1989*; *Dawe et al., 2007*; *Ishikawa and Marshall, 2011*; *Avasthi and Marshall, 2012*; *Chemes, 2012*). The symmetry and diameter of centrioles (and thereby of flagella/cilia) is, to a large extent, established through scaffolding by the centriolar cartwheel, a structure with a central ring-like hub and nine radially projecting spokes that contact the peripheral centriolar microtubules (*Azimzadeh and Marshall, 2010*; *Brito et al., 2012*; *Gonczy, 2012*).

The cartwheel is organized through the homo-oligomerization of the highly conserved centriolar protein SAS-6 (*Nakazawa et al., 2007*; *Kitagawa et al., 2011*; *van Breugel et al., 2011*). High-resolution crystal structures of zebrafish and *C. reinhardtii* SAS-6 fragments together with biochemical and biophysical characterizations (*Kitagawa et al., 2011*; *van Breugel et al., 2011*) demonstrated that two dimerization interfaces in SAS-6 mediate this oligomerization: a coiled-coil domain that forms a rod-like parallel dimer; and a globular N-terminal domain that forms a curved head-to-head dimer. Modeling these two dimer interactions together in silico resulted in ring assemblies that were compatible with the symmetry and dimension of cartwheels observed in vivo. In these models, the N-terminal head domains constitute the cartwheel hubs from which the coiled-coil rods project to form the cartwheel spokes (*Kitagawa et al., 2011*; *van Breugel et al., 2011*). However, so far, no high-resolution structure of the SAS-6 cartwheel is available. Although rotary shadowing EM studies suggested that SAS-6 might be able to form cartwheels, the available resolution was insufficient to determine the symmetry of the observed assemblies directly (*Kitagawa et al., 2011*). Another, higher resolution, EM study with recombinant SAS-6 revealed cartwheel-like assemblies with an eightfold, not a ninefold symmetry

*For correspondence: vanbreug@mrc-lmb.cam.ac.uk

[†]Present address: Novartis Institutes for BioMedical Research, Basel, Switzerland

Competing interests: The authors declare that no competing interests exist.

**eLife digest** Many cells have tiny hair-like structures called cilia on their surface that are important for communicating with other cells and for detecting changes in the cell's surroundings. Some cilia also beat to move fluids across the cell surface—for example, to move mucus out of the lungs—or act as flagella that undergo rapid whip-like movements to propel cells along. Cilia are formed when a small cylindrical structure in the cell called a centriole docks against the cell membrane and subsequently grows out. However, many of the details of this process are poorly understood.

One of the earliest events in centriole assembly is the formation of a central structure that looks like a cartwheel. This cartwheel acts as a scaffold onto which the rest of the centriole is then added. It has been proposed that a protein called SAS-6 can build this cartwheel just by interacting with itself. However, this has so far not been shown clearly. Now, using a technique called X-ray crystallography, van Breugel et al. directly confirm this hypothesis. This is significant because it demonstrates that the simple self interaction of a protein could lie at the heart of building a complex structure like a centriole.

The single-celled human parasites that spread diseases such as Leishmaniasis, Chagas disease, and sleeping sickness rely on flagella to move around and interact with their surroundings. If SAS-6 cannot assemble into the cartwheel structure, flagella cannot form correctly, potentially stopping the parasites. By screening a library of small molecules, van Breugel et al. found one that partially disrupted the interactions of SAS-6 with itself in the test tube. This small molecule interacted only very weakly with SAS-6 and was not specific for SAS-6 from the disease-causing organism. These unfavourable properties therefore make this compound of no immediate use. However, this result nevertheless shows that small molecules can impair SAS-6 function at least in the test tube and that the development of a more efficient inhibitor might therefore be possible.

(*van Breugel et al., 2011*), while a third EM study showed the presence of SAS-6 tetramers (*Gopalakrishnan et al., 2010*). Furthermore, the available biochemical and biophysical data do not provide evidence for efficient cartwheel formation by SAS-6 in solution (*Gopalakrishnan et al., 2010*; *Kitagawa et al., 2011*; *van Breugel et al., 2011*), raising the question of whether SAS-6 alone is sufficient to organize ninefold symmetric cartwheels and, if so, under what conditions.

SAS-6 is a highly conserved protein that is found in all eukaryotes that have cilia/flagella during some of their life-cycle stages (*Carvalho-Santos et al., 2010*; *Hodges et al., 2010*). Inhibiting SAS-6 self-assembly by point mutations in vivo abolishes the formation of centrioles (*Kitagawa et al., 2011*; *van Breugel et al., 2011*; *Lettman et al., 2013*) and thereby flagella (*van Breugel et al., 2011*). Thus, targeting SAS-6 oligomerization by inhibitors could be a strategy to disable flagellogenesis in organisms that cause human disease and rely on flagella for their pathogenicity. The Trypanosomatids are of special interest in this respect. They consist of parasitic, eukaryotic protozoa with a single flagellum and comprise members that cause major human diseases, such as sleeping sickness (*Trypanosoma brucei*), Chagas disease (*Trypanosoma cruzi*), and Leishmaniasis (*Leishmania spec.*) (*Simpson et al., 2006*). These organisms display complex life cycles in which they shuttle between an insect vector and human or animal hosts. Their flagellum provides key roles in this life cycle through its multiple functions in essential cellular processes such as motility, signalling, sensing, and attachment (*Vaughan and Gull, 2003*; *Ralston et al., 2009*; *Gluenz et al., 2010*). Furthermore, the flagellar pocket, a membrane invagination from which the flagellum emerges, is the exclusive site of vesicular membrane traffic in these organisms (*Overath et al., 1997*; *Landfear and Ignatushchenko, 2001*; *Field and Carrington, 2004*, *2009*; *Overath and Engstler, 2004*). It is therefore not surprising that flagella have been demonstrated or are proposed to be key factors in the pathogenicity of these organisms (*Vaughan and Gull, 2003*; *Ralston et al., 2009*; *Gluenz et al., 2010*).

As in other eukaryotes, the flagellum of the Trypanosomatids is templated from centrioles (basal bodies). Tomographic EM studies demonstrated the presence of canonical, ninefold symmetric cartwheel structures in the centre of their centrioles (*Lacomble et al., 2009*). Currently, no high-resolution structures of SAS-6 homologues from Trypanosomatids are available, and it is unclear whether oligomerization of their SAS-6 homologues could be inhibited.

## Results

### *Leishmania major* SAS-6 is highly similar to other SAS-6 homologues

The genomes of *Trypanosoma brucei*, *Trypanosoma cruzi*, and *Leishmania major* have recently been sequenced (*Berriman et al., 2005*; *El-Sayed et al., 2005*; *Ivens et al., 2005*). BLAST searches identified the likely SAS-6 homologues in these organisms with similar domain architectures (*Figure 1A*, *Figure 1—figure supplement 1*) and sequence identities to human SAS-6 of 21.0 ± 1.4% in 459 ± 21 aligned residues. Multiple sequence alignment of their N-terminal domains (*Figure 1—figure supplement 1B*) shows that key residues are well conserved compared to zebrafish SAS-6, the closest homologue of human SAS-6 for which high-resolution structures are available (*van Breugel et al., 2011*). Different from zebrafish SAS-6, they have long N-terminal extensions (*Figure 1A*, *Figure 1—figure supplement 1A,B*) that vary in length and are poorly conserved. Since these extensions hindered our crystallization attempts, we largely removed them in the constructs used in this manuscript. The part of the extensions still present did not show electron density in our crystal structures.

To elucidate the structural organization of *L. major* SAS-6, we solved the structure of its N-terminal domain (Lm SAS-6$^{97–274}$) by X-ray crystallography to a resolution of 2.2 Å (*Table 1*, *Table 2*; *Figure 1B*). The asymmetric unit (ASU) of the crystal contained three molecules that were virtually identical to each other (139 ± 2 selected pairs superpose with an rmsd of 0.70 ± 0.13 Å in secondary structure matching). Our structure demonstrates a high similarity of *L. major* SAS-6 to previously solved SAS-6 structures (*Kitagawa et al., 2011*; *van Breugel et al., 2011*) (secondary structure matching to the *D. rerio* SAS-6 head domain results in an rmsd of 1.59 ± 0.09 Å with 133 ± 2 selected pairs). Like these, the N-terminal domain of *L. major* SAS-6 consists of a 7-stranded β-barrel, capped by a helix-turn-helix motif, that forms a curved cross-handshake homo-dimer within the crystal through a highly conserved interaction interface (*Figure 1B*, *Figure 1—figure supplement 1B,C*). In this dimer, the β-hairpins formed by β-strands β6 and β7 pack antiparallelly against each other. Phenylalanine 257 at the tip of this hairpin is inserted into a conserved hydrophobic pocket that is constituted by the helix-turn-helix motif and the base of this hairpin in the B-C homo-dimer (formed by chain B and chain C in the crystal, *Figure 1B*). Dimerization is also observed in solution, as judged by equilibrium ultracentrifugation, and then depends on the presence of F257 (*Figure 1D*). The measured $K_D$ of this dimerization is ~600 μM and therefore approximately 5- to 10-fold weaker than seen for other SAS-6 homologues (*Kitagawa et al., 2011*; *van Breugel et al., 2011*).

To ascertain the role of the coiled-coil domain of *L. major* SAS-6, we tried to crystallize constructs that included both the N-terminal domain and parts of the coiled-coiled domain, but initially failed to obtain diffraction-grade crystals. However, by introducing the F257E mutation to strongly weaken head-to-head dimerization we managed to crystallize construct Lm SAS-6$^{97–320}$ F257E that contained the N-terminal head domain and the first seven heptad-repeats of the coiled-coil domain. We subsequently solved its X-ray structure to a resolution of 2.9 Å (*Table 1*; *Figure 1E*). The asymmetric unit of the crystal contained four molecules that were highly similar to each other and superposed with an rmsd of 1.17 ± 0.50 Å in secondary structure matching with 165 ± 16 selected pairs. The crystal structure revealed that the *L. major* SAS-6 coiled-coil domain is a parallel dimer and packs via conserved interactions against the N-terminal head-domains as seen in other SAS-6 homologues (*Figure 1E*, *Figure 1—figure supplement 1D*). Thus, our structural analyses demonstrate that *L. major* SAS-6 is highly similar to other SAS-6 homologues.

### Alternative dimerization arrangements of *Leishmania major* SAS-6 reveal a dimerization-impaired state

Our structural analysis also revealed the presence of an alternative arrangement of the head-to-head homo-dimer of *L. major* SAS-6. In the Lm SAS-6$^{97–274}$ crystal, the A–A homo-dimer (formed by chain A and symmetry-related chain A) shares the features of the B–C homo-dimer described above. However, in the A–A homo-dimer, the Y215 sidechain is flipped into its own hydrophobic pocket resulting in the displacement from this pocket of residue F257 of its homo-dimer partner (*Figure 1C*) probably weakening the interaction. Intriguingly, we found a similar arrangement in the Lm SAS-6$^{97–320}$ F257E crystal. The F257E mutation abolishes head-to-head dimerization in solution (*Figure 1D*), yet, in the crystal, due to the high protein and precipitant concentrations, some of the SAS-6 molecules present are found nevertheless in head-to-head dimers and form a curved octamer. In this octamer residues Y215 are also swung into their own hydrophobic pockets, while E257 point away from them

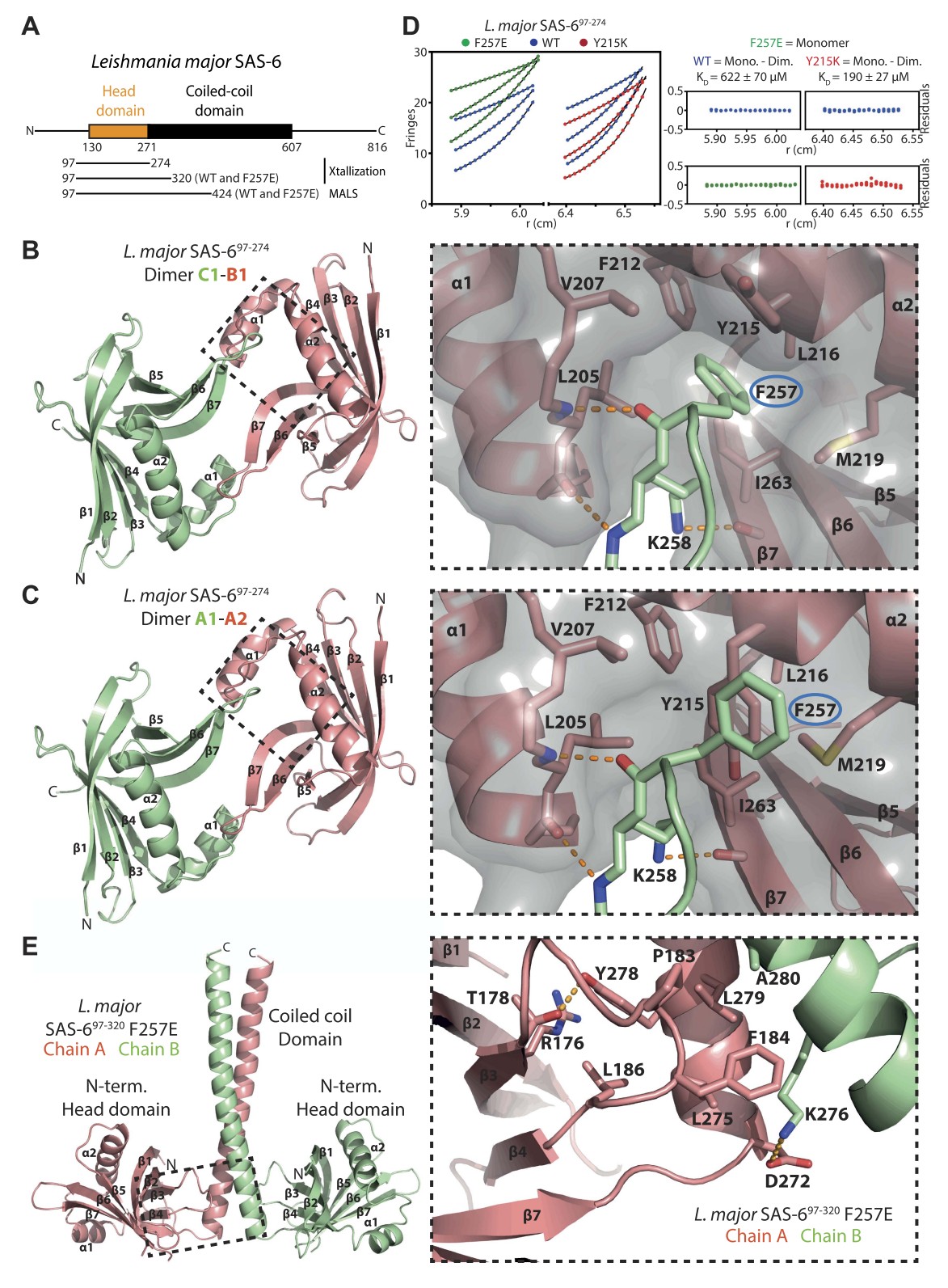

**Figure 1**. Structural and biophysical characterization of *L. major* SAS-6. (**A**) Domain overview of *L. major* SAS-6. Lines indicate constructs that were used in this work. (**B** and **C**) Left: ribbon presentation of the head-to-head dimers of *L. major* SAS-6's N-terminal domain present in the SAS-6$^{97–274}$ crystal. Shown are the dimers formed between chain B and chain C (**B**) and chain A and symmetry-related chain A (**C**). α-helices (α) and β-sheets (β) are numbered sequentially. Right: detailed views of the corresponding dimerization interfaces. Interface residues are labelled and shown in sticks, dotted orange lines indicate
*Figure 1. Continued on next page*

*Figure 1. Continued*

hydrogen bonds. The two dimers show largely identical side-chain orientations in their interfaces. Note, however, that F257 (ringed in blue) in the B–C dimer inserts into a hydrophobic pocket, while in the A–A dimer Y215 is flipped into this pocket and displaces F257. To better illustrate this, a semi-transparent molecular surface of one of the subunits is also presented (grey). (**D**) Sedimentation-equilibrium analytical ultracentrifugation data for 400 µM *L. major* SAS-6$^{97–274}$ wild-type (blue circles) and F257E mutant (green circles) and Y215K mutant (red circles) obtained at 11300, 17000, and 21200 rpm. Data for the F257E mutant were fitted to an ideal single-species model (solid line). Analysis of multiple concentrations gave a molecular weight of 17,727 ± 219 Da, close to the expected molecular weight for the monomer of 19,640 Da. As initial fits to a similar model for the WT and Y215K data gave higher molecular weights of 27,589 ± 209 Da and 30,951 ± 595 respectively, the data were fitted to a monomer–dimer equilibrium model (solid line) giving dissociation constants, $K_D$, of 622 ± 70 µM for the WT and 190 ± 27 µM for Y215K mutant. The plots on the right show the residuals of the fits to the data for the wild-type (blue circles) and the corresponding F257E mutant (green circles) and Y215K mutant (red circles). (**E**) Left: ribbon presentation of the *L. major* SAS-6$^{97–320}$ F257E coiled-coil dimer structure (chain A: red, chain B: green). Right: detailed view of the region boxed on the left. Interaction interface between N-terminal head domain and the coiled-coil stalk. Residues that make contact are labelled and are shown as sticks, dotted orange lines indicate hydrogen bonds.

The following figure supplements are available for figure 1:

**Figure supplement 1**. *L. major* SAS-6 and *Danio rerio* SAS-6 are highly similar.

**Figure supplement 2**. The closed Y215 conformation is observed in the low affinity head-to-head homo-dimer of the F257E mutant.

(***Figure 1—figure supplement 2***). These data suggest that the closed Y215 conformation corresponds to a low-affinity dimerization state of the head domains and might therefore constitute a potential regulatory mechanism of SAS-6 oligomerization. In solution, the presence of a dimerization-impaired state could also explain the relatively low affinity of head-to-head dimerization of wild-type *L. major*

**Table 1.** Native dataset analysis and refinement statistics

| | *L. major* SAS-6$^{97–274}$ **WT** | *L. major* SAS-6$^{97–320}$ **F257E** | *L. major* SAS-6$^{97–320}$ **WT** |
|---|---|---|---|
| Beamline | Diamond I04 | ESRF ID29 | ESRF BM14 |
| Space Group | P43212 | C121 | H3 |
| Wavelength (Å) | 0.9794 | 0.90 | 0.97813 |
| Monomers in the asymmetric unit | 3 | 4 | 6 |
| Unit Cell dimensions (Å) | a = 84.25 b = 84.25 c = 239.94 α = 90.0 β = 90.0 γ = 90.0 | a = 108.9 b = 81.25 c = 133.1 α = 90.0 β = 91.5 γ = 90.0 | a = 482.7 b = 482.7 c = 43.13 α = 90.0 β = 90.0 γ = 120.0 |
| Resolution (Å) | 48.9–2.2 | 46.91–2.9/3.4 (anisotropy) | 66.9–3.5/4.2 (anisotropy) |
| Completeness (overall/inner/outer shell) | 99.9/98.9/100 | 99.9/99.1/99.9 | 100/99.1/100 |
| Rmerge (overall/inner/outer shell) | 0.144/0.064/1.505 | 0.152/0.032/2.570 | 0.335/0.074/3.061 |
| Rpim (overall/inner/outer shell) | 0.065/0.030/0.657 | 0.061/0.013/1.033 | 0.073/0.016/0.704 |
| Mean I/σI (overall/inner/outer shell) | 7.6/18.6/1.4 | 10.6/41.4/0.9 | 8.2/33.3/1.5 |
| Multiplicity (overall/inner/outer shell) | 5.9/5.5/6.0 | 7.2/6.7/7.1 | 22.3/21.9/19.8 |
| Number of reflections | 45,569 | 24,090 | 48,521 |
| Number of atoms | 3536 | 5653 | 8777 |
| Waters | 141 | 0 | 0 |
| Rwork/Rfree (% data used) | 20.8/24.5 (5.0%) | 23.7/25.6 (5.0%) | 22.4/24.2 (5.1%) |
| rmsd from ideal values: bond length/angles | 0.010/1.349 | 0.008/1.255 | 0.006/1.348 |
| Mean B value | 51.5 | 92.8 | 150.6 |
| Average Real-space correlation coefficient | 0.976 | 0.924 | 0.881 |
| Molprobity Score | 1.14 (100$^{th}$ percentile) | 1.33 (100$^{th}$ percentile) | 1.41 (100$^{th}$ percentile) |

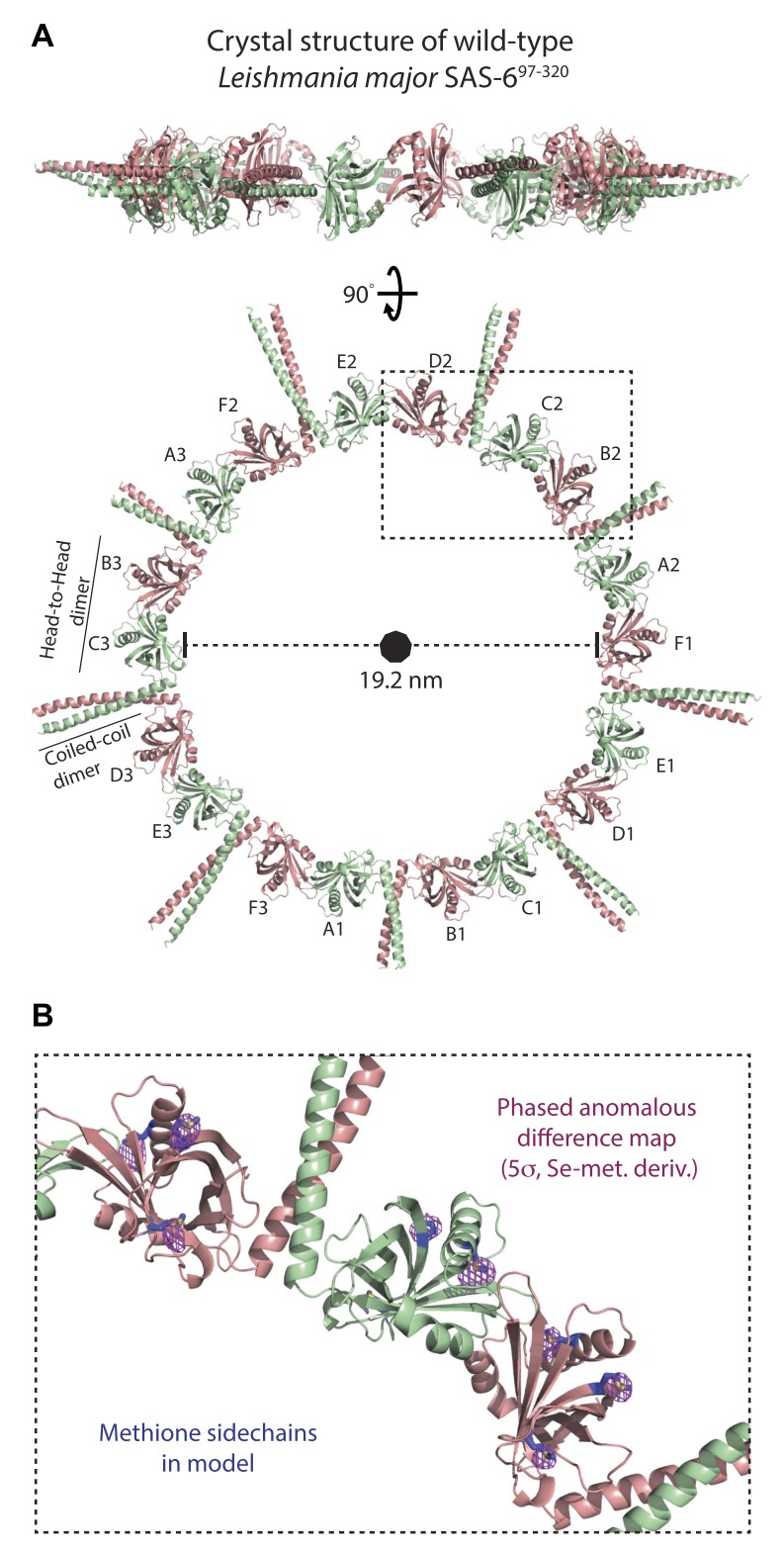

**A** Crystal structure of wild-type
*Leishmania major* SAS-6$^{97\text{-}320}$

90°

19.2 nm

Head-to-Head dimer

Coiled-coil dimer

**B**

Phased anomalous
difference map
(5σ, Se-met. deriv.)

Methione sidechains
in model

**Figure 2**. *L. major* SAS-6$^{97–320}$ crystallizes as a ninefold symmetric ring with dimensions similar to those of centriolar cartwheels observed in vivo. (**A**) Ribbon presentation of the *L. major* SAS-6$^{97–320}$ structure. Shown is the ring assembly present in the unit cell of the *L. major* SAS-6$^{97–320}$ crystal. Protein chains are colored alternatingly in green and red to allow easier comparison with *Figure 1*. Top: side-view, bottom: face-on view of the *L. major* SAS-6$^{97–320}$
*Figure 2. Continued on next page*

SAS-6 apparent in analytical ultracentrifugation (~600 µM compared to 50–100 µM observed for other species) since other SAS-6 homologues do not have aromatic residues at the equivalent position of Y215 that could play such a role.

To test whether the Y215 closed conformation significantly compromises dimerization in solution, we mutated Y215 in Lm SAS-6$^{97–274}$ to lysine that is unable to block the hydrophobic pocket in a similar way. Subsequently, we subjected the purified protein to analytical ultracentrifugation (*Figure 1D*). The measured $K_D$ of head-to-head dimerization of the Y215K mutant was ~200 µM and therefore approximately threefold lower than for the corresponding wild-type protein. We conclude that Y215 acts to partially inhibit head-to-head dimerization of *L. major* SAS-6.

### *Leishmania major* SAS-6 crystallizes as ninefold symmetric rings that are highly similar to centriolar cartwheels in vivo

The presence of curved SAS-6 octamers in the crystal of the Lm SAS-6$^{97–320}$ F257E mutant that is strongly impaired in its ability to form head-to-head dimers suggests that wild-type versions of SAS-6 would adopt even larger assemblies. The low affinity of head-to-head dimerization together with the concomitant sample heterogeneity makes EM studies of the resulting assemblies technically challenging. To overcome these limitations we tried to crystallize *L. major* SAS-6 constructs with both dimerization interfaces intact. Obtaining crystals that diffracted well enough to solve their structure proved difficult. However, with the wild-type construct of Lm SAS-6$^{97–320}$, we finally succeeded to find a crystal form of the space-group H3 that diffracted to a resolution of ~3.5 Å along the l-axis with anisotropy limiting the resolution along the h-k plane to ~4.2 Å. Using the structure of the Lm SAS-6$^{97–274}$ B–C homo-dimer as a search model, a molecular replacement solution could be found that allowed the subsequent placement of the coiled-coil part present in the construct (*Figure 2*). The unit cell of the Lm SAS-6$^{97–320}$ crystal form contained three 9-fold symmetric SAS-6 rings with three SAS-6 dimers constituting the ASU. These three dimers were highly similar to each other and overlayed with an rmsd of 0.88 ± 0.29 Å in secondary structure matching with 357 ± 14 selected pairs. No electron density was seen inside the SAS-6 rings. The inner diameters of the SAS-6 rings are ~19 nm and correspond well to the diameters of cartwheel hubs observed in vivo (*Lacomble et al., 2009*; *Guichard et al., 2010, 2012*). In the crystal,

*Figure 2. Continued*

ring structure. The nonagon in the center of the ring indicates the (quasi-) ninefold symmetry axis. The ASU of the crystal contained six SAS-6 monomers that are labelled from A–F. No clear electron density could be seen for the distal part of the coiled-coil of the A–B dimer, probably due to the lack of stabilizing crystal packing interactions compared to the C–D and E–F dimer (*Figure 2—figure supplement 1B*). (**B**) Detailed view of the region boxed in (**A**). Shown in blue sticks are the methionine-side chains of the SAS-6$^{97–320}$ model. In magenta, iso-mesh representation of the phased anomalous difference map at a contour level of σ = 5 showing the selenium positions in the crystallized selenomethionine derivate of *L. major* SAS-6$^{97–320}$.

The following figure supplements are available for figure 2:

**Figure supplement 1**. The crystal packing interactions observed in the *wild-type L. major* SAS-6$^{97–320}$ crystal.

**Figure supplement 2**. The interfaces critical for ring formation in the *wild-type L. major* SAS-6$^{97–320}$ crystal are similar in orientation to those observed in the SAS-6$^{97–320}$ F257E and the SAS-6$^{97–274}$ crystals.

SAS-6 rings are stacked onto each other. Neighbouring rings interact through inter-digitation of their coiled-coil domains in an antiparallel way (*Figure 2—figure supplement 1*). We confirmed our structural model by calculating a phased anomalous map using the refined phases based on our model and the amplitudes of an isomorphic dataset collected from crystals of the selenomethionine derivative (*Figure 2B*; *Table 3*). Peaks in this map correlated with the positions of methionines in our model. Thus, under appropriate conditions, *L. major* SAS-6 can adopt ninefold symmetric rings that are highly similar to cartwheels observed in vivo.

## A small compound can inhibit SAS-6 oligomerization in vitro

To find out whether we could inhibit SAS-6 oligomerization, we conducted a small-scale fragment screen using a custom library of halogenated fragments (HEFLib) (*Wilcken et al., 2012*). First, pools of compounds were screened for their ability to bind to and thereby cause a shift perturbation in the {$^1$H,$^{15}$N}-HSQC NMR spectrum of $^{15}$N labelled Lm SAS-6. To avoid ambiguities in the interpretation of shift perturbations that could stem from the partial presence of oligomers, we used the Lm SAS-6$^{97–274}$ F257E mutant for these binding studies that is monomeric in solution (*Figure 1D*). One dimensional spin-echo experiments provided a crude estimate of 18 ms for the backbone amide $^1$H $T_2$ relaxation time constants, consistent with the molecular mass of the construct (~20 kDa) (*Anglister et al., 1993*).

To determine the putative interaction sites of binding candidates, we assigned the backbone resonances of Lm SAS-6$^{97–274}$ F257E using $^{13}$C,$^{15}$N double-labelled protein and mapped the HSQC chemical shift perturbations onto the crystal structure of wild-type Lm SAS-6$^{97–274}$. Strong perturbations in chemical shifts are most consistent with compound PK9119 ((5-bromo-7-ethyl-1*H*-indol-3-ylmethyl)-dimethyl-amine, *Figure 3A*) binding adjacent to the head-to-head dimerization interface of Lm SAS-6$^{97–274}$ (*Figure 3B*, *Figure 3—figure supplement 1A,B*). Smaller, but significant shift changes indicate that binding may alter this interface by affecting the conformation of the helix-turn-helix motif that constitutes a part of it. We also examined aromatic side chain $^1$H resonances of the Phe and Tyr residues using Cβ–Hδ correlation maps, which enabled us to identify two side-chains (F199, F212) that are perturbed on binding of PK9119 (*Figure 3B*, *Figure 3—figure supplement 1C*). These two side-chains cluster together around the helix-turn-helix motif. HSQC chemical shift titration experiments with PK9119 and *L. major* SAS-6$^{97–274}$ F257E suggest a millimolar binding affinity; the low solubility of PK9119 in aqueous solutions and lack of a reference compound for competition binding assays makes more accurate $K_D$ determinations technically challenging.

To determine whether PK9119 affects oligomerization of *L. major* SAS-6, we subjected wild-type and F257E mutant protein to size-exclusion chromatography—multi-angle light scattering (SEC-MALS) in the presence or absence of 1 mM PK9119 (*Figure 3C*). Since we were unable to make full length *L. major* SAS-6 recombinantly, we used a *L. major* SAS-6 construct that contained the N-terminal domain and approximately half of its coiled-coil domain (Lm SAS-6$^{97–424}$) for this assay. Similar to the previous findings (*Gopalakrishnan et al., 2010*; *Kitagawa et al., 2011*; *van Breugel et al., 2011*), we did not find evidence for a stable ring-fraction of SAS-6 in solution, but found a complex equilibrium of SAS-6 oligomers (ranging up to approximately SAS-6 hexamers) for the wild-type construct in the absence of PK9119, while the F257E mutant was a stable dimer under these conditions, in agreement

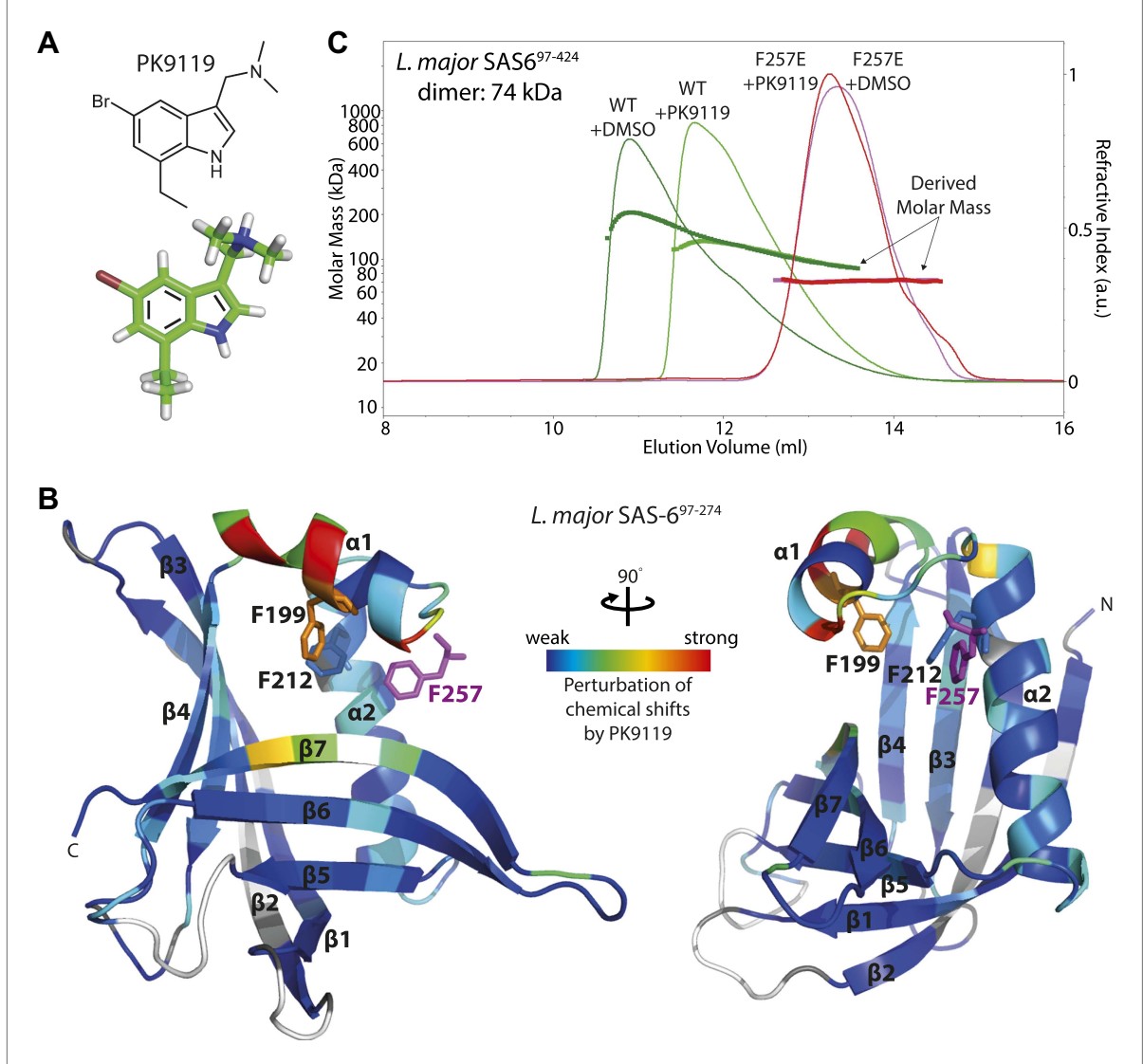

**Figure 3**. A small chemical compound can inhibit SAS-6 oligomerization. (**A**) The chemical structure of compound PK9119 as a structure formula (top) or three-dimensional model (bottom). (**B**) Heat map of the chemical shift perturbations in the {$^{1}$H-$^{15}$N}-HSQC spectrum of $^{15}$N-labelled *L. major* SAS-6$^{97-274}$ F257E in the presence of 2 mM PK9119. Data are plotted onto the crystal structure of wild-type *L. major* SAS-6$^{97-274}$. Higher shift perturbation is depicted in warmer colors, whilst prolines (not observable in the HSQC) are colored grey, and unassigned/untraced residues are colored white. The magenta F257 is from the homo-dimer partner and is inserted into the hydrophobic pocket of the dimerization interface. Note that the chemical shift perturbations cluster close to this pocket. Side-chains are drawn for F199 and F212 that showed robust perturbations in (HB)CB(CGCD)HD correlation spectra in the presence of 1 mM PK9119 (*Figure 3—figure supplement 1C*). (**C**) SEC-MALS chromatogram of *L. major* SAS-6$^{97-424}$ showing the refractive index signal with the derived molar masses indicated by the thicker horizontal lines. *L. major* SAS-6$^{97-424}$ displayed a distribution of masses from that of the dimer up to >200 kDa consistent with a concentration driven self-association equilibrium. In the presence of 1 mM PK9119 the maximal mass was almost halved. The F257E mutant displayed a constant mass of 71 kDa in the absence and presence of PK9119 consistent with the mass of a dimer of *L. major* SAS-6$^{97-424}$. All samples were injected on SEC-MALS at 25 mg/ml (675 µM in monomer). Due to dilution during SEC the peak concentrations achieved were a factor of ~10 lower than this (~68 µM in monomer).

The following figure supplements are available for figure 3:

**Figure supplement 1**. Chemical shift perturbation of SAS-6 by PK9119.

**Figure supplement 2**. PK9119 affects the oligomerization of zebrafish SAS-6.

with stable dimer formation through its coiled-coil domain. When the runs were repeated in the presence of PK9119, we saw a clear shift in the elution volume for the wild-type but not the F257E SAS-6 construct towards smaller molecular weights and a decrease in the analysed mass from MALS, demonstrating that PK9119 partially affects head-to-head dimerization. We also repeated this experiment with the equivalent constructs of zebrafish SAS-6 (Dr SAS-6$^{1–326}$, wild-type and F131D) under similar conditions. The results showed that zebrafish SAS-6 oligomerization was also affected by the presence of PK9119, although to a lesser extent than *L. major* SAS-6 oligomerization (*Figure 3—figure supplement 2*). Thus, PK9119 appears to be a general inhibitor of SAS-6 oligomerization in vitro with some preference for the *L. major* variant, and represents a starting point for further screening of chemical analogs or fragment evolution.

## Discussion

SAS-6 organizes the rotationally ninefold symmetric cartwheel (*Nakazawa et al., 2007*; *Kitagawa et al., 2011*; *van Breugel et al., 2011*), an early assembly intermediate of centrioles that participates in establishing their symmetry and diameter (*Nakazawa et al., 2007*; *Brito et al., 2012*; *Gonczy, 2012*). Based on high-resolution structures of SAS-6 fragments and in silico modeling, detailed models of how SAS-6 self-associates to organize these cartwheels have been proposed (*Nakazawa et al., 2007*; *Kitagawa et al., 2011*; *van Breugel et al., 2011*). However, these models have so far not been confirmed; EM studies with recombinant SAS-6 constructs were either of too low resolution (*Kitagawa et al., 2011*) or showed assemblies that were not ninefold symmetric (*Gopalakrishnan et al., 2010*; *van Breugel et al., 2011*). Furthermore, several studies found no evidence for complete ring-formation by recombinant SAS-6 in solution (*Gopalakrishnan et al., 2010*; *Kitagawa et al., 2011*; *van Breugel et al., 2011*). Thus, it has been unclear if and under what conditions SAS-6 is sufficient to assemble ninefold symmetric cartwheels and thereby assist in dictating centriole symmetry.

We were unable to detect cartwheel formation by recombinant *L. major* SAS-6 in solution, confirming these previous findings. However, we demonstrate that under high protein and precipitant concentrations, a *L. major* SAS-6 construct crystallized as ninefold symmetric rings with a diameter very similar to cartwheel hubs in vivo; this provides the first unambiguous experimental evidence that SAS-6 is able to form ninefold symmetric cartwheels on its own. Our results suggest that SAS-6 needs to be highly concentrated locally in order to be able to form cartwheels. Intriguingly, in vivo, SAS-6 is indeed highly enriched at the site of centriole formation (*Kleylein-Sohn et al., 2007*; *Strnad et al., 2007*; *Dammermann et al., 2008*; *Blachon et al., 2009*; *Sonnen et al., 2012*; *Lettman et al., 2013*). However, in vivo, SAS-6 assembly occurs in the context of other essential centriole duplication factors, some of which can be found in a complex with SAS-6 (*Stevens et al., 2010*; *Tang et al., 2011*; *Lin et al., 2013*). It is therefore likely that additional centriole proteins assist SAS-6 assembly, especially in light of the fact that several studies (*Gopalakrishnan et al., 2010*; *Kitagawa et al., 2011*; *van Breugel et al., 2011*), and our study presented here, have been unable to demonstrate efficient cartwheel assembly by SAS-6 in solution.

Why does SAS-6 not form cartwheels efficiently? The interaction interfaces that are critical for ring formation (i.e., the head-to-head dimer interface and the head-domain–coiled-coil interface) are both relatively small (*Figures 1, 2*; *Kitagawa et al., 2011*; *van Breugel et al., 2011*). In solution, SAS-6 oligomers are therefore likely to show some 'wobble' that would make ring closure inefficient and assembled rings unstable. However, in our ring structure, both of these critical interfaces are similar in their orientations to those observed in the crystals of the Lm SAS-6$^{97–320}$ F257E and the Lm SAS-6$^{97–274}$ constructs that did not crystallize as rings (*Figure 2—figure supplement 2*). Thus, although clearly not occurring efficiently in solution, our data suggest that ninefold symmetric ring formation might correspond to a weakly favored SAS-6 conformation that needs to be stabilized. In our crystal structure, rings are stabilized by crystal packing interactions and, in vivo, probably by SAS-6 interacting proteins. This model could explain why SAS-6 is required for the faithful establishment of centriole symmetry, while not being the sole determinant of their ninefold symmetry in vivo (*Nakazawa et al., 2007*).

In basal bodies, cartwheels are stacked onto each other with a vertical distance between cartwheel hubs of ~8 nm (*Guichard et al., 2012*). Although we observed a similar stacking in our cartwheel hub structure (*Figure 2—figure supplement 1*), the corresponding distance was only ~4 nm. Furthermore, although a comparable packing was previously observed in a crystal of the N-terminal head domain of zebrafish SAS-6 (*van Breugel et al., 2011*), the underlying packing interactions are based on a non-conserved interface and are therefore unlikely to be of relevance in vivo. Recent EM tomograms of

basal bodies from Trichonympha (*Guichard et al., 2012*, *2013*) rather suggest that ring stacking in vivo is based on parallel interactions between the SAS-6 coiled-coil stalks and vertical interactions of SAS-6 associated components at the periphery of centriolar cartwheels.

Interestingly, we discovered a dimerization-impaired state of *Leishmania major* SAS-6 that could provide a potential regulatory mechanism for its assembly. In this state, residue Y215 blocks the hydrophobic pocket into which F257 from its homo-dimer partner binds. Mutating Y215 to lysine resulted in an apparent threefold increase in dimerization affinity as measured by analytical ultracentrifugation. If open and closed Y215 conformations are in equilibrium with each other, the dimerization affinity of the closed Y215 state in solution would be even lower than the measured value, since, in this case, ultracentrifugation analyses the association of a mixture of open and closed states. We speculate that a release of this closed SAS-6 state through either a regulatory protein or a direct phosphorylation of Y215 could provide a simple mechanism to trigger SAS-6 assembly locally by increasing the concentration of assembly efficient SAS-6. Should this mechanism indeed be used in vivo (which is currently unknown), it would probably be confined to the Trypanosomatids as other SAS-6 homologues appear not to have tyrosine/aromatic residues in the position equivalent to Y215 that could play an equivalent role.

Finally, as a proof-of-principle we show that oligomerization of SAS-6 can be inhibited by the small molecule PK9119 in vitro. The evolutionary conservation of the hydrophobic pocket involved in dimerization together with PK9119's relatively broad activity in inhibiting both *L. major* and (to a lesser extent) *D. rerio* SAS-6 could suggest that PK9119 targets this pocket. However, our NMR binding studies are most consistent with PK9119 binding adjacent to the head-to-head dimerization interface and thereby altering its structure subtly (*Figure 3B*). Confirming this notion, when we modeled the SAS-6–PK9119 complex with the HADDOCK software package (available at http://haddock.science. uu.nl/services/HADDOCK/haddock.php) (*de Vries et al., 2010*; *Wassenaar et al., 2012*), using high ambiguity restraints derived from the observed chemical shift perturbations in HSQC spectra (6 restraints >0.2 ppm), we found that of the 200 docked structures 98% occupied a single cluster in which PK9119 was bound to helix α1, but not on the side lining the hydrophobic pocket, but on its opposite side, facing away from this pocket (data not shown). Clearly, an elucidation of a high-resolution structure of PK9119 bound to its target would be important to understand how exactly PK9119 functions in inhibiting *L. major* SAS-6. The binding affinity of our compound is currently low (in the mM range) and it does not show strong species specificity. To establish whether PK9119 has any potential to be improved in these critical aspects, it will be essential to systematically explore chemical modifications of its central indole scaffold. Regardless of these limitations, our demonstration that SAS-6 can be inhibited in vitro is a first, small step towards the goal of developing SAS-6 inhibitors that also could be used in vivo, for example as a cell-biological tool.

## Materials and methods

### Recombinant protein expression and purification

All *L. major* constructs were made synthetically as codon-optimized genes (IDT, Coralville, Iowa). Zebrafish SAS-6 constructs were described earlier (*van Breugel et al., 2011*). All constructs were N-terminally His-tagged. Proteins were expressed in *E. coli* BL21 Rosetta and purified using standard methods via NiNTA (Qiagen, Hilden, Germany) chromatography, proteolytic tag cleavage, size-exclusion chromatography and ion-exchange chromatography. The *L. major* SAS-6$^{97-274}$ and SAS-6$^{97-320}$ selenomethionine derivatives were purified in the same way, but expression was in M9 medium supplemented with 2 mM MgSO$_4$, 0.4% (wt/vol) glucose, 25 μg/ml FeSO$_4$.7H$_2$O, 40 μg/ml amino acid mix (excluding Methionine), 1 μg/ml riboflavin, 1 μg/ml niacinamide, 0.1 μg/ml pyridoxine monohydrochloride, 1 μg/ml thiamine and 40 μg/ml seleno-L-methionine. All purified *L. major* constructs included the extra sequence GP, zebrafish SAS-6 GPH at their N-termini from the cloning/protease cleavage site.

### Crystallization

SeMet *L. major* SAS-6$^{97-274}$ crystals were obtained using the sitting drop method with a reservoir solution of 100 mM bisTris pH 5.1, 200 mM MgCl$_2$, 20% (wt/vol) PEG-3350 at 16°C. Drops were set up using 100 nl protein solution and 100 nl of reservoir solution. After half a day, the crystals were mounted in 100 mM bisTris pH 5.1, 200 mM MgCl$_2$, 10% (wt/vol) PEG-3350, 25% (wt/vol) Glycerol and flash-frozen in liquid nitrogen.

**Table 2.** SeMet *L. major* SAS-6[97–274] WT dataset analysis

| Beamline | Diamond I03 | | |
|---|---|---|---|
| Space group | P43212 | | |
| Wavelength (Å) | 0.9794 (Peak) | 0.9796 (Inflection) | 0.9393 (Remote) |
| Unit Cell dimensions (Å) | a = 84.19 b = 84.19 c = 239.6 α = 90.0 β = 90.0 γ = 90.0 | a = 84.24 b = 84.24 c = 239.7 α = 90.0 β = 90.0 γ = 90.0 | a = 84.22 b = 84.22 c = 239.7 α = 90.0 β = 90.0 γ = 90.0 |
| Resolution (Å) | 68.9–2.3 | 68.9–2.3 | 68.9–2.3 |
| Completeness (overall/inner/outer shell) | 99.8/99.9/99.7 | 99.7/99.9/99.7 | 99.8/99.7/99.7 |
| Rmerge (overall/inner/outer shell) | 0.130/0.066/0.942 | 0.122/0.048/1.001 | 0.131/0.052/0.973 |
| Rpim (overall/inner/outer shell) | 0.066/0.036/0.467 | 0.062/0.026/0.497 | 0.066/0.028/0.486 |
| Mean I/sd(I) (overall/inner/outer shell) | 7.4/18.8/1.7 | 7.8/21.8/1.7 | 7.2/20.2/1.6 |
| Multiplicity (overall/inner/outer shell) | 4.8/4.1/4.9 | 4.7/4.1/4.9 | 4.7/4.0/4.8 |
| Se sites found/expected | 14/12 | | |

Native *L. major* SAS-6[97–274] was crystallized at 16°C using the sitting drop method with 200 nl of the protein solution and 200 nl of the reservoir solution (100 mM bisTris pH 5.1, 200 mM MgCl$_2$, 20% [wt/vol] PEG-3350). The crystals were mounted in 100 mM bisTris pH 5.1, 200 mM MgCl$_2$, 10% (wt/vol) PEG-3350, 25% (wt/vol) glycerol after 1 day and flash-frozen in liquid nitrogen.

*L. major* SAS-6[97–320] F257E was crystallized in sitting drops in 100 mM NaCitrate pH 5.85, 21% (wt/vol) PEG-3000 at 16°C using 1 µl of protein solution and 1 µl of the reservoir solution. The crystals were mounted after 4–5 days in 100 mM NaCitrate pH 5.85, 21% (wt/vol) PEG-3000 and increasing amounts of PEG-400 to a final concentration of 20% (wt/vol) before flash-freezing them in liquid nitrogen.

*L. major* SAS-6[97–320] WT crystals were obtained using the sitting drop method with a reservoir solution of 100 mM bisTris pH 6.23, 100 mM NaAcetate (not pH adjusted), 26.5–27% (wt/vol) PEG-400 at 16°C. Drops were set up using 1.8 µl of protein solution and 1.8 µl of the reservoir solution. After 3 days, the crystals were mounted in 100 mM bisTris pH 6.23, 100 mM NaAcetate (not pH adjusted), 28% (wt/vol) PEG-400.

**Table 3.** SeMet *L. major* SAS-6[97–320] WT dataset analysis

| Beamline | ESRF BM14 |
|---|---|
| Space group | H3 |
| Wavelength (Å) | 0.97872 (Peak) |
| Unit Cell dimensions (Å) | a = 481.7 b = 481.7 c = 42.9 α = 90.0 β = 90.0 γ = 120.0 |
| Resolution (Å) | 48.2–4.0/5.4 (anisotropy) |
| Completeness (overall/inner/outer shell) | 100.0/98.8/100.0 |
| Rmerge (overall/inner/outer shell) | 0.194/0.054/2.118 |
| Rpim (overall/inner/outer shell) | 0.091/0.026/0.987 |
| Mean I/sd(I) (overall/inner/outer shell) | 3.8/13.7/0.9 |
| Multiplicity (overall/inner/outer shell) | 5.6/5.5/5.6 |

SeMet *L. major* SAS-6[97–320] WT crystals were obtained using the sitting drop method with a reservoir solution of 100 mM bisTris pH 6.23, 100 mM NaAcetate (not pH adjusted), 25.5% (wt/vol) PEG-400 at 16°C. Drops were set up using 1.5 µl of protein solution and 1.5 µl of the reservoir solution. After 3 days, the crystals were mounted in 100 mM bisTris pH 6.23, 100 mM NaAcetate (not pH adjusted), 30% (wt/vol) PEG-400.

The protein concentrations of the crystallized constructs were determined by the Bradford assay with BSA as a standard and were: 80.3 mg/ml (SeMet *L. major* SAS-6[97–274]), 29.3 mg/ml (*L. major* SAS-6[97–274]), 94.7 mg/ml (*L. major* SAS-6[97–320] F257E), 50.3 mg/ml SeMet *L. major* SAS-6[97–320] and 50.1 mg/ml (*L. major* SAS-6[97–320]).

## Data collection and processing

Data sets were integrated and scaled using MOSFLM (*Leslie and Powell, 2007*) (SeMet and native *L. major* SAS-6[97–274], *L. major* SAS-6[97–320]) or XDS (*Kabsch, 2010*) (*L. major* SAS-6[97–320] F257E and SeMet *L. major* SAS-6[97–320]). Data sets were scaled using SCALA or AIMLESS (*Evans, 2006*; *Evans and Murshudov, 2013*). The *L. major* SAS-6[97–274] structure was solved from the

corresponding 3-wavelength SeMet dataset by MAD using the SHELX CDE pipeline in HKL2MAP (*Pape and Schneider, 2004*), resulting in clear electron density into which an initial model was built using BUCANNEER (*Cowtan, 2006*, *2008*) and manual building. REFMAC (*Murshudov et al., 2011*) was used to refine the model against the native data set with manual building done in Coot (*Emsley and Cowtan, 2004*). *L. major* SAS-6$^{97–320}$ WT and F257E were solved by molecular replacement in Phaser (*McCoy et al., 2007*) using the *L. major* SAS-6$^{97–274}$ structure as a search model (SAS-6$^{97–274}$ monomer (F257E) or the BC-dimer (WT). The models were subsequently further built in Coot (*Emsley and Cowtan, 2004*) and refined in REFMAC (*Murshudov et al., 2011*) and Phenix.refine (*Afonine et al., 2005*) using NCS and (for WT *L. major* SAS-6$^{97–320}$) TLS refinement with separate TLS groups for the globular N-terminal and the coiled-coil domains and also using as a reference model restraint the B chain of *L. major* SAS-6$^{97–274}$ (residue 130–271). Refinement yielded clear density for the missing coiled-coil part of these constructs.

## Analytical ultracentrifugation

Equilibrium sedimentation experiments were performed on an Optima XL-I analytical ultracentrifuge (Beckmann, Brea, California) using An50Ti rotors. Sample volumes of 110 µl with protein concentrations of 100, 200, and 400 µM were loaded in 12 mm 6-sector cells and centrifuged at 11300, 17000, and 21200 rpm until equilibrium was reached at 4°C. At each speed, comparison of several scans was used to judge whether or not equilibrium had been reached. Buffer conditions were 50 mM Tris, 100 mM NaCl, pH 8.0. The solvent density and viscosity ($\rho$ = 1.00557 g/ml and $\eta$ = 1.6056 mPa·s) were calculated using Sednterp (Dr Thomas Laue, University of New Hampshire, Sednterp server available at: http://sednterp.unh.edu. Desktop version can be downloaded from: http://bitcwiki.sr.unh.edu/index.php/Downloads). Data were processed and analysed using UltraSpin software (available at: http://www.mrc-lmb.cam.ac.uk/dbv/ultraspin2/) and SEDPHAT (*Schuck, 2003*).

## NMR experiments

Small molecules were screened for binding to approximately 40 µM $^{15}$N-labelled protein in aqueous phosphate buffer (25 mM Phosphate, 150 mM NaCl, 2 mM DTT, pH 7.2) and up to 2 mM ligand concentration, with a total of 5% (vol/vol) DMSO-$d_6$. {$^{1}$H-$^{15}$N}-fast-HSQC spectra (*Mori et al., 1995*) were recorded using a Bruker Avance spectrometer operating at 800 MHz $^{1}$H frequency, with a 5 mm cryogenic inverse probe and sample temperature of 298 K. The digital resolution of the processed data was 3.2 and 5.3 Hz/point in $f_2$ and $f_1$, respectively. Backbone resonance assignments were obtained from HNCACB, CBCA(CO)NH, HN(CA)CO, HNCO and HNCANH spectra at 600 MHz $^{1}$H frequency, acquired using unmodified Bruker pulse programs and a protein concentration of 400 µM. Aromatic sidechain resonances were assigned from a (HB)CB(CGCD)HD spectrum. Data were processed using TopSpin version 3 (commercially available from Bruker, Billerica, Massachusetts, details available at http://www.bruker.com/products/mr/nmr/nmr-software/software/topspin/) and analysed using Sparky (Goddard & Kneller, UCSF, San Francisco, available at http://www.cgl.ucsf.edu/home/sparky/).

## HADDOCK calculations

Models of the complex between SAS-6 and PK9119 were generated by submitting the crystal structure coordinates of a *L. major* SAS-6$^{97–274}$ monomer to the WeNMR server (available at: https://www.wenmr.eu) (*de Vries et al., 2010*; *Wassenaar et al., 2012*), using default HADDOCK parameters and CNS topology parameters for PK9119 based on the PRODRG predictions provided on the HADDOCK server (available at http://haddock.science.uu.nl/services/HADDOCK/haddock.php).

## Size exclusion chromatography coupled to multi-angle light scattering (SEC-MALS)

The mass in solution of *L. major* SAS-6$^{97–424}$, wild-type and F257E mutant, Dr SAS-6$^{1–326}$, wild-type and F131D mutant, was determined by SEC-MALS measurements using a Wyatt Heleos II 18 angle light scattering instrument coupled to a Wyatt Optilab rEX online refractive index detector. Detector 12 in the Heleos instrument was replaced with Wyatt's QELS detector for dynamic light scattering measurement. Protein samples (100 µl) were resolved on a Superdex S-200 10/300 analytical gel filtration column (GE Healthcare, Little Chalfont, UK) running at 0.5 ml/min in 50 mM bisTris, 100 mM NaCl, pH 7.0 buffer, containing 0.126% (vol/vol) DMSO and ±1 mM chemical compound PK9119 ((5-bromo-7-ethyl-1H-indol-3-ylmethyl)-dimethyl-amine, Sigma-Aldrich, St. Louis, Missouri) before passing through the light scattering and refractive index detectors in a standard SEC-MALS format.

Buffers were filtered through a 0.22-µm filter before usage to remove any PK9119 precipitates. Protein concentration was determined from the excess differential refractive index based on 0.186 RI increment for 1 g/ml protein solution. The concentration and the observed scattered intensity at each point in the chromatograms were used to calculate the absolute molecular mass from the intercept of the Debye plot using Zimm's model as implemented in Wyatt's ASTRA software (commercially available from Wyatt technology, Santa Barbara, California, details at: http://www.wyatt.com/products/software/astra.html).

## Acknowledgements

For beamline support, MvB would like to acknowledge Dr Carina Loblex (I04) and Dr James Nicholson (I03) at Diamond Light Source, Oxford, UK, and Dr Philippe Carpentier (ID29) and Dr Hassan Belrhali, Dr Babu Manjasetty (BM14) at the European Synchrotron Radiation Facility (ESRF), Grenoble, France. We thank Dr Stefan Freund (MRC-LMB, Cambridge, UK) for assistance with NMR data acquisition and discussion. The structures presented in this work have been deposited under pdb codes 4ckm, 4ckn and 4ckp.

## Additional information

### Funding

| Funder | Grant reference number | Author |
|---|---|---|
| Medical Research Council | MC_UP_1201/03 | Mark van Breugel |

The funder had no role in study design, data collection and interpretation, or the decision to submit the work for publication.

### Author contributions

MB, Crystallized the constructs and solved their X-ray structures, Contributed to writing the manuscript; RW, TJR, Performed the NMR experiments, Contributed to writing the manuscript; SHM, Carried out the analytical ultracentrifugation, Contributed to writing the manuscript; CMJ, Performed the SEC-MALS experiments, Contributed to writing the manuscript

## Additional files

### Major datasets

The following datasets were generated:

| Author(s) | Year | Dataset title | Dataset ID and/or URL | Database, license, and accessibility information |
|---|---|---|---|---|
| van Breugel M | 2012 | Structure of the N-terminal domain of Leishmania SAS-6 | 4CKM; http://www.rcsb.org/pdb/search/structidSearch.do?structureId=4ckm | Publicly available at the RCSB Protein Data Bank (http://www.rcsb.org/). |
| van Breugel M | 2013 | Structure of an N-terminal fragment of Leishmania SAS-6 containing parts of its coiled coil domain, F257E mutant | 4CKN; http://www.rcsb.org/pdb/search/structidSearch.do?structureId=4ckn | Publicly available at the RCSB Protein Data Bank (http://www.rcsb.org/). |
| van Breugel M | 2013 | Structure of an N-terminal fragment of Leishmania SAS-6 that contains part of its coiled coil domain | 4CKP; http://www.rcsb.org/pdb/search/structidSearch.do?structureId=4ckp | Publicly available at the RCSB Protein Data Bank (http://www.rcsb.org/). |

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
