## [Decision Letter]

Thank you for sending your work entitled “Structure of the SAS-6 cartwheel hub from *Leishmania major*” for consideration at *eLife*. Your article has been favorably evaluated by a Senior editor, John Kuriyan, and 3 reviewers, one of whom, Erich Nigg, has agreed to reveal his identity.

The Senior editor and reviewers discussed their comments before we reached this decision, and the Senior editor has assembled the following comments to help you prepare a revised submission.

The centriolar protein SAS-6 has recently been shown to be a major determinant of the 9-fold symmetry of the cartwheel, an important structure in centriole assembly. (This was shown independently by two groups, one being the author of the present study). This conclusion was based on a combination of X-ray structure data, in silico modeling and rotatory shadowing EM. Although immediately embraced by the community as an important and beautiful advance, it had not previously been established formally that SAS-6 alone can assemble into rings with 9-fold symmetry.

The gap in structural knowledge concerning SAS-6 has been filled in the present study. Structures of fragments containing the N-terminal head domain (97-274) or a dimerization deficient head domain + coiled coil fragment (97-320 F237E) provide new insights into the interactions that stabilize head-to-head SAS-6 dimer interfaces and the interface between the coiled-coil and the head domain, respectively. Most importantly, the authors were also able to solve the structure of the wild-type 97-320 construct, and made the important observation that this fragment crystallizes in a nine-fold symmetry, creating a ring with a diameter very similar to cartwheel hubs observed in centrioles in vivo. Thus, the authors' data show that SAS-6 is sufficient to form a ring with nine-fold symmetry and that other components are not required.

The authors also provide evidence for a potential auto-inhibitory conformation in *Leishmania* SAS-6 (involving Y215) – however, this aromatic residue is not present in animal SAS-6 and its in vivo relevance is uncertain. In addition, the manuscript identifies a pocket at a dimer interface between head domains that can be targeted by small molecules. Although they emphasize the potential of future therapeutic uses of such compounds, the present data are rather preliminary and can only be seen as a first modest step in this direction (the compound is of low affinity and shows little species-specificity).

In conclusion, that the authors have determined a crystal structure of the cartwheel is really quite remarkable, given its large diameter and the lack of stabilizing internal spokes. The additional structures add to the strength of the story. The identification of a potential small molecule inhibitor is nice and makes it a sound story for publication. The auto-inhibitory discussion may be a little overdone with little experimental support, but it seems reasonable to present the idea that Y215 may play a direct role in SAS-6 assembly regulation. Thus, the paper should be published in *eLife* once the authors have addressed the following minor comments, using their own judgement as to how best to respond to each of these points. The revised paper will receive a very rapid final decision by the editor without further review, and so a clear discussion of how these points are addressed will be helpful.

Minor comments:

1) The authors state that SAS-6 rings are stacked onto each other in the crystal and they also refer to literature stating that Sas-6 rings are stacked onto each other in basal bodies (26). They should discuss these data, and also a recent publication reporting that *C. elegans* SAS-6 forms a spiral (Hilbert et al., PNAS 2013).

2) A couple of the reviewers had issues related to arguments about energy.

The full architecture could be more stable than intermediates; however, if it were low energy than it would be favored in solution, which it is not. It is also clear that it isn't stable in any in vitro context outside the crystal. Since it is difficult to argue about free energy or energy from crystal structures, particularly due to the confounding effects of local concentration, the authors may wish to just exclude these arguments (the structure speaks for itself). Specifically, in the section entitled ‘*Leishmania major* SAS-6 crystallizes as 9-fold symmetric rings that are highly similar to centriolar cartwheels in vivo’: the authors argue that because each of the dimers in the asymmetric unit in the 97-320 structure has a similar conformation, this conformation represents a low energy state of SAS-6. The logic of this argument and the rationale for making it are unclear. Any conformation that crystallizes (including tetrameric and octameric SAS-6) is likely to be a low energy conformation. The observation that under the 97-320 crystallization conditions SAS-6 forms a nine-membered ring is not sufficient evidence to claim the head-to-head dimer conformation in the nine-fold symmetrical ring is more stable in general than the dimer interfaces previously observed in octameric or tetrameric structures.

3) The authors state in the Abstract that their data indicate how cartwheel assembly is regulated in vivo. While the observation that a residue in Lm SAS-6 can adopt a conformation that weakens head-to-head contacts is suggestive, it is premature to state that this observation provides the structural basis for regulation of cartwheel-imposed symmetry.

4) Section entitled ‘A small compound can inhibit SAS-6 oligomerization in vitro’: The NMR data show perturbations in and around the hydrophobic dimerization pocket when the inhibitor binds. It would be easier to assess the correlation between the chemical shift data and the residues in the dimerization pocket if the authors explicitly listed the residues (and ideally, show them in the figure) that are within interaction distance of F527 versus those that show a perturbation. Because of the sensitivity of chemical shift perturbations to allosteric structural changes, the authors should consider modifying the statement “Strong perturbations indicate that PK9119 … binds” to indicate that the perturbation data is consistent with PK9119 binding to the hydrophobic pocket.

5) Discussion section: The authors observe that the nine-membered rings stack vertically in the crystal based on interactions between the head domains, with a stacking distance of about 4 nm. That the ring stacking distance is much greater in vivo and that the stacking interfaces in the crystal are not conserved argues against the authors' speculation that the observed ring stacking contacts in the crystal are important during cartwheel assembly. It would be an improvement to remove this speculation.

6) Last paragraph of Discussion section: The authors note that PK9119 can be docked into the hydrophobic pocket. The authors do not present criteria for evaluating whether a particular ligand can be docked or not. Therefore, the authors should either include the docking data in the Results section, making the details of the modeling and it's interpretation clear, or this statement should be removed from the Discussion.

7) Results section” The term 'helix-turn-helix motif' is defined as an architecture in certain DNA binding proteins. I found the use of the phrase here inappropriate. If it truly is a helix-turn-helix motif it should be explained better.

8) The description of the B-C homo-dimer is not symmetry related, unless there is a misunderstanding. This is an asymmetric unit dimer while the A-A homo-dimer is symmetry related. It should be clarified. The use of color in Figure 1 is confusing as the molecules are colored the same even though different interfaces are being described. Altering the color to make these more distinct would be helpful. Specifically, using different colors for the subunit interfaces would help in clarity as one relates the complexes to each other. Minimally, flipping the coloring in E would make it easier to understand how the dimerization interfaces differ. This could then be echoed in Figure 2. Adding numbers to D would help more than these traces.

9) Section entitled ‘Alternative dimerization arrangements of *Leishmania major* SAS-6 suggest an auto-inhibitory mechanism’: This is where the auto-inhibitory discussion becomes confusing. Auto-inhibitory suggests that there must be a state change to get binding. Instead, what is described is that the conformation of Y215 lowers the affinity but doesn't inhibit binding. This can be described better.

10) “This map showed a good agreement...” is strange phrasing. Did the methionine positions agree with the anomalous map? If so, it should just say 'Peaks in this map correlated to positions of methionines in our model.'

11) Discussion section: The argument that concentration mediates cartwheel formation seems overstated. As the assembly process requires other proteins, it seems more likely that other proteins convey stability to form the complex.

12) Last paragraph of Discussion section: If the authors would like to discuss PK9119 docking they should describe this in more detail in the results. The methods do not give enough details to suggest that there is any reason to trust the results. This should be either dropped or expanded.

13) Regarding the docking, the argument that 'our results suggest that targeting SAS-6 oligomerization...to interfere with pathogenicity' may be a bit too strong. Clearly, others have demonstrated oligomerization of SAS-6 homologues is critical in centriole formation and flagella are important in tyrpanosomatids, so the idea is not a novel one. That said, the authors should more strongly state that they have demonstrated that a small molecule can affect this assembly in vitro. However, that is a long way from showing that this interface will be a good target in vivo, moreover, it isn't obvious that they have identified a great lead compound for this.

---

## [Author Response]

*1) The authors state that SAS-6 rings are stacked onto each other in the crystal and they also refer to literature stating that Sas-6 rings are stacked onto each other in basal bodies (*[26]*). They should discuss these data, and also a recent publication reporting that C. elegans SAS-6 forms a spiral (Hilbert et al., PNAS 2013)*.

And

*5) Discussion section: The authors observe that the nine-membered rings stack vertically in the crystal based on interactions between the head domains, with a stacking distance of about 4 nm. That the ring stacking distance is much greater* in vivo *and that the stacking interfaces in the crystal are not conserved argues against the authors' speculation that the observed ring stacking contacts in the crystal are important during cartwheel assembly. It would be an improvement to remove this speculation*.

We agree that our speculation that the ring stacking observed in the crystal might serve in vivo to facilitate/template further ring formation might be seen as too far stretched. Although we had clearly discussed the shortcomings of this hypothesis previously, we now completely rewrote this part of the discussion to point out that the observed ring stacking is probably not relevant in vivo. We also further expanded our discussion of how ring stacking might be mediated in vivo in light of the recent EM tomographic reconstructions of basal bodies. Concerning the Hilbert et al., (PNAS 2013) publication reporting that *C. elegans* SAS-6 might form 9-fold symmetric spirals we preferred not to expand the Discussion (which is already long) further. Although Hilbert et al.’s data demonstrate that SAS-6 self-assembly could also be compatible with a different organization (spiral vs ring) Hilbert et al show that this would be a nematode-specific phenomenon since a nematode-specific residue sterically makes ring-formation in *C. elegans* unlikely. So far, SAS-6 spirals have not been demonstrated in *C. elegans* in vivo and we therefore believe that tying in their findings in our discussion would not be straightforward and would run the risk of distracting from the “canonical” SAS-6 homologues with well-established cartwheel geometries.

*2) A couple of the reviewers had issues related to arguments about energy*.

*The full architecture could be more stable than intermediates; however, if it were low energy than it would be favored in solution, which it is not. It is also clear that it isn't stable in any* in vitro *context outside the crystal. Since it is difficult to argue about free energy or energy from crystal structures, particularly due to the confounding effects of local concentration, the authors may wish to just exclude these arguments (the structure speaks for itself). Specifically, in the section entitled ‘Leishmania major SAS-6 crystallizes as 9-fold symmetric rings that are highly similar to centriolar cartwheels* in vivo*’: the authors argue that because each of the dimers in the asymmetric unit in the 97-320 structure has a similar conformation, this conformation represents a low energy state of SAS-6. The logic of this argument and the rationale for making it are unclear. Any conformation that crystallizes (including tetrameric and octameric SAS-6) is likely to be a low energy conformation. The observation that under the 97-320 crystallization conditions SAS-6 forms a nine-membered ring is not sufficient evidence to claim the head-to-head dimer conformation in the nine-fold symmetrical ring is more stable in general than the dimer interfaces previously observed in octameric or tetrameric structures*.

We have been unclear in our argument and therefore corrected the corresponding sections of our manuscript. However, we maintain that the nine-membered ring might correspond to a weakly favored conformation. We show now with a supplementary figure (Figure 2—figure supplement 2) that the head-to-head orientation and the relative orientation of head-domains to coiled coil stalks in the nine-membered ring are similar to those observed in the two other crystal structures presented in our manuscript that do not crystallize as rings (97-320 FE construct and WT 97-274 construct). Furthermore, in previous studies, in silico modeling of 9-fold symmetric rings required only small changes of the crystal structures of zebrafish and *Chlamydomonas* SAS-6 fragments (van Breugel M et al. Science. 2011; 331(6021):1196-9, Kitagawa et al. Cell. 2011; 144(3):364-375). To our mind these data suggest that despite some “wobble”, due to the small interaction interfaces involved, the observed orientations in our ring structure might very well be weakly favored. Efficient ring-formation is clearly not observed in solution, probably due to this flexibility and also due to the low affinities involved. Under our assay conditions (SEC-MALS) it would be difficult to observe a small fraction of nine-fold symmetric rings present in solution. We argue that SAS-6 rings need to be stabilized, either by crystal packing interactions (as in our structure in vitro) or by additional SAS-6 interacting proteins (in vivo). We believe that this model could explain why SAS-6 is essential for faithful establishment of the centriolar symmetry, while not being the sole determinant of it. We discuss this now in detail in our manuscript.

*3) The authors state in the Abstract that their data indicate how cartwheel assembly is regulated* in vivo*. While the observation that a residue in Lm SAS-6 can adopt a conformation that weakens head-to-head contacts is suggestive, it is premature to state that this observation provides the structural basis for regulation of cartwheel-imposed symmetry*.

In our manuscript we previously made it clear that it is entirely unknown whether this mechanism is indeed used in vivo or not. Since we want to avoid any misunderstanding we further weakened or removed corresponding statements in our manuscript.

*4) Section entitled ‘A small compound can inhibit SAS-6 oligomerization* in vitro*’: The NMR data show perturbations in and around the hydrophobic dimerization pocket when the inhibitor binds. It would be easier to assess the correlation between the chemical shift data and the residues in the dimerization pocket if the authors explicitly listed the residues (and ideally, show them in the figure) that are within interaction distance of F527 versus those that show a perturbation. Because of the sensitivity of chemical shift perturbations to allosteric structural changes, the authors should consider modifying the statement “Strong perturbations indicate that PK9119 … binds” to indicate that the perturbation data is consistent with PK9119 binding to the hydrophobic pocket*.

And

*6) Last paragraph of Discussion section: The authors note that PK9119 can be docked into the hydrophobic pocket. The authors do not present criteria for evaluating whether a particular ligand can be docked or not. Therefore, the authors should either include the docking data in the Results section, making the details of the modeling and it's interpretation clear, or this statement should be removed from the Discussion*.

And

*12) Last paragraph of Discussion section: If the authors would like to discuss PK9119 docking they should describe this in more detail in the results. The methods do not give enough details to suggest that there is any reason to trust the results. This should be either dropped or expanded*.

Our NMR data (Figure 3) show that PK9119 causes the largest shift perturbations in helix α1 that lines the hydrophobic pocket where F257 is normally inserted. However, the extent of the perturbations are largest on the outside of this helix not the side facing the pocket, which can be clearly evaluated from Figure 3. Furthermore, we looked at aromatic side chain 1H resonances of Phe and Tyr residues using Cβ-Hδ correlation maps, and found that F199, F212 (both lining the hydrophobic pocket) are perturbed on binding of PK9119 (Figure 3—figure supplement 1). Together, the NMR data therefore would probably be most consistent with PK9119 binding outside of the hydrophobic pocket in a cleft on the other side of the hydrophobic pocket divider, although interpretations are difficult due to sidechain reorganisation and other conformational changes that can contribute to shift perturbation. We had tried to be clear in our manuscript that PK9119 likely binds around (or in) the hydrophobic pocket. PK9119 docking/binding to the pocket was only mentioned in the Discussion as a possibility.

The reviewers’ comments made us realize that we need to clarify these issues further. We modeled the complex between SAS-6 and PK9119 using the HADDOCK docking package, applying restraints derived from the observed chemical shift perturbations. We found that, indeed, the ligand consistently docks close to helix α1, but not on the side lining the hydrophobic pocket, but on its opposite side, facing away from this pocket. However, since there is considerable variation in the binding mode of the ligand in the 200 trial structures, we have preferred not to include an image of the HADDOCK output that might be misconstrued as representing a true solution structure. Thus, in our revised manuscript, we now do not include any docking figure, as this would distract from the NMR data. Furthermore, we clarified each statement in the manuscript that refers to the potential PK9119 binding site to reflect what is discussed here.

*7) Results section” The term 'helix-turn-helix motif' is defined as an architecture in certain DNA binding proteins. I found the use of the phrase here inappropriate. If it truly is a helix-turn-helix motif it should be explained better*.

The helix-turn-helix motif is a functionally divergent motif that in some proteins has lost the DNA binding function and acquired other functions such as protein-protein interaction for instance (Aravind L et al., FEMS Microbiol Rev. 2005 Apr;29(2):231-62.). XLF and XRCC4, the structural homologues of SAS-6 that are involved in DNA repair, both have helix-turn-helix motifs (Junop MS et al. EMBO J. 2000 Nov 15;19(22):5962-70. Li Y et al. EMBO J. 2008 Jan 9;27(1):290-300.) similar to SAS-6’s (van Breugel M et al. Science. 2011 Mar 4;331(6021):1196-9) (it is at this stage uncertain whether the HTH motifs of XRCC4 and XLF have a role in DNA binding). Thus, it seems to be appropriate to call the equivalent motif in SAS-6 a helix-turn-helix motif. As SAS-6 is not involved in DNA binding, we furthermore believe that it will not be confusing to leave the corresponding passages as they are.

*8) The description of the B-C homo-dimer is not symmetry related, unless there is a misunderstanding. This is an asymmetric unit dimer while the A-A homo-dimer is symmetry related. It should be clarified. The use of color in*
Figure 1
*is confusing as the molecules are colored the same even though different interfaces are being described. Altering the color to make these more distinct would be helpful. Specifically, using different colors for the subunit interfaces would help in clarity as one relates the complexes to each other. Minimally, flipping the coloring in E would make it easier to understand how the dimerization interfaces differ. This could then be echoed in*
Figure 2*. Adding numbers to D would help more than these traces*.

We reformatted the ASU to avoid the misunderstanding and changed the corresponding descriptions. We flipped the colors of the subunits in Figure 1 to facilitate the appreciation of how SAS-6 oligomerization occurs and kept this coloring scheme also in Figure 2 to make the two figures color-wise directly comparable. Concerning the ultracentrifugation panel, we added the K_D_’s to Figure 1, but also kept the traces as we believe it is good to see the quality of the fit.

*9) Section entitled ‘Alternative dimerization arrangements of Leishmania major SAS-6 suggest an auto-inhibitory mechanism’: This is where the auto-inhibitory discussion becomes confusing. Auto-inhibitory suggests that there must be a state change to get binding. Instead, what is described is that the conformation of Y215 lowers the affinity but doesn't inhibit binding. This can be described better*.

It is not clear to us that auto-inhibition as a term must necessarily be limited to a state of complete self-inhibition. Furthermore, it is unclear to what extent the closed conformation inhibits oligomerization. Since open and closed state are likely to be in equilibrium with each other (change of state requires only a rotamer change of Y215) we measure in ultracentrifugation a mixture of these states and thus end up with an “average” K_D_. Depending where the equilibrium lies it is thus very well possible that the closed state has a very low ability to dimerize in solution. In agreement, the F257E mutant oligomerizes in the closed state but only in the crystal and not in solution arguing that the remaining contact interface indeed mediates only a very weak interaction. Nevertheless, at this stage we are unable to directly demonstrate the K_D_ of the closed state alone. Furthermore, at least in the crystal, the closed state is indeed still able to oligomerize, potentially leading to misunderstandings. Thus, we follow the advide of the reviewers, avoid calling it auto-inihibition, and describe/discuss this part of our manuscript better and in more detail making all these points clearer.

*10) “This map showed a good agreement...” is strange phrasing. Did the methionine positions agree with the anomalous map? If so, it should just say 'Peaks in this map correlated to positions of methionines in our model.*'

The methionine positions agree very well with the anomalous map (see Figure 2) and we corrected our phrasing as suggested.

*11) Discussion section: The argument that concentration mediates cartwheel formation seems overstated. As the assembly process requires other proteins, it seems more likely that other proteins convey stability to form the complex*.

We do not think that one argument excludes the other. Our data suggest that SAS-6 can – under high concentrations – form 9-fold symmetric rings and the published data suggest that SAS-6 is indeed highly enriched at the place of centriole duplication. However, SAS-6 ring formation clearly requires stabilization, either by crystal packing interactions (in vitro) or by other SAS-6 interacting proteins (in vivo). Thus, we argue that centriole assembly needs both high SAS-6 concentrations and SAS-6 interacting proteins that stabilize a 9-fold symmetric assembly. We further clarified our discussion of this point in our manuscript.

*13) Regarding the docking, the argument that 'our results suggest that targeting SAS-6 oligomerization...to interfere with pathogenicity' may be a bit too strong. Clearly, others have demonstrated oligomerization of SAS-6 homologues is critical in centriole formation and flagella are important in tyrpanosomatids, so the idea is not a novel one. That said, the authors should more strongly state that they have demonstrated that a small molecule can affect this assembly* in vitro*. However, that is a long way from showing that this interface will be a good target* in vivo*, moreover, it isn't obvious that they have identified a great lead compound for this*.

By no means do we want to get across as having identified a great lead or that SAS-6 is necessarily a fantastic target for inhibition by small molecules in vivo. In fact, we were very careful in clearly emphasizing the shortcomings of the compound both in terms of affinity as well as in terms of species specificity (and solubility). As the reviewers pointed out, what we achieved is only a proof-of-principle that SAS-6 oligomerisation can be inhibited in vitro by a small molecule. Thus, we changed/toned down further all parts of the manuscript where there still might be misunderstandings about these points.